# Progression in the Management of Non-Idiopathic Pulmonary Fibrosis Interstitial Lung Diseases, Where Are We Now and Where We Would Like to Be

**DOI:** 10.3390/jcm10061330

**Published:** 2021-03-23

**Authors:** Tinne Goos, Laurens J. De Sadeleer, Jonas Yserbyt, Geert M. Verleden, Marie Vermant, Stijn E. Verleden, Wim A. Wuyts

**Affiliations:** 1Laboratory of Respiratory Diseases and Thoracic Surgery (BREATHE), Department CHROMETA, KU Leuven, B-3000 Leuven, Belgium; tinne.goos@kuleuven.be (T.G.); laurens.desadeleer@kuleuven.be (L.J.D.S.); jonas.yserbyt@uzleuven.be (J.Y.); geert.verleden@uzleuven.be (G.M.V.); marie.vermant@uzleuven.be (M.V.); stijn.verleden@kuleuven.be (S.E.V.); 2Department of Respiratory Diseases, University Hospitals Leuven, B-3000 Leuven, Belgium

**Keywords:** progressive fibrosing interstitial lung disease, pulmonary fibrosis, interstitial lung disease, management

## Abstract

A significant proportion of patients with interstitial lung disease (ILD) may develop a progressive fibrosing phenotype characterized by worsening of symptoms and pulmonary function, progressive fibrosis on chest computed tomography and increased mortality. The clinical course in these patients mimics the relentless progressiveness of idiopathic pulmonary fibrosis (IPF). Common pathophysiological mechanisms such as a shared genetic susceptibility and a common downstream pathway—self-sustaining fibroproliferation—support the concept of a progressive fibrosing phenotype, which is applicable to a broad range of non-IPF ILDs. While antifibrotic drugs became the standard of care in IPF, immunosuppressive agents are still the mainstay of treatment in non-IPF fibrosing ILD (F-ILD). However, recently, randomized placebo-controlled trials have demonstrated the efficacy and safety of antifibrotic treatment in systemic sclerosis-associated F-ILD and a broad range of F-ILDs with a progressive phenotype. This review summarizes the current pharmacological management and highlights the unmet needs in patients with non-IPF ILD.

## 1. Introduction

In addition to idiopathic pulmonary fibrosis (IPF), which is defined as a relentless progressive disease, a significant proportion of patients with other interstitial lung disease (ILD) subtypes may develop a progressive phenotype characterized by worsening of symptoms, deterioration of pulmonary function, progressive pulmonary fibrosis on chest computed tomography (CT) and increased mortality. Other non-IPF ILD subtypes that may be associated with a progressive fibrosing phenotype include connective tissue disease-associated ILD (CTD-ILD), idiopathic non-specific interstitial pneumonia (iNSIP), sarcoidosis, hypersensitivity pneumonitis (HP), occupational ILD and unclassifiable ILD (uILD) [1,2,3,4]. Based on survey data and cohort studies, it is estimated that 30% of non-IPF ILD patients will develop a progressive fibrosing phenotype [5,6,7]. In this review, we summarize the current pharmacological management and highlight the unmet needs in patients with non-IPF ILD.

## 2. Historical Context

For decades, the mainstay of treatment in IPF was immunosuppressive therapy, based on the hypothesis that inflammation leads to injury and fibrosis [8]. Thereafter, it was thought that an oxidant–antioxidant imbalance may contribute to this process of fibrosis in IPF, leading to the IFIGENIA-trial in 2005 [9]. The major merit of this trial was that it made the field realize that larger randomized controlled trials (RCTs) in ILD were possible. The management of IPF changed drastically in 2012, as the PANTHER-IPF trial showed that immunosuppressive treatment resulted in increased mortality [10]. Therefore, it became clear that immunosuppressive agents provided no benefit and, on the contrary, were harmful in IPF patients. This observation supported the new hypothesis that IPF was not the result of an exaggerated immune response but should be considered as an aberrant progressive self-sustaining fibrotic process initiated by repeated epithelial injury without an identifiable cause [8]. Indeed, two antifibrotic drugs, pirfenidone and nintedanib, were shown to be effective in IPF by reducing forced vital capacity (FVC) decline and are currently recommended for the treatment of IPF [11,12,13,14].

While antifibrotic drugs became the standard of care in IPF, immunosuppressive agents are still the mainstay of treatment in non-IPF fibrosing ILD (F-ILD). However, it became clear that a subset of patients with non-IPF F-ILD continued to have progressive fibrosis despite conventional immunosuppressive treatment and had an IPF-like outcome [1,15,16,17,18]. Additionally, from a pathophysiological perspective, studies have reported shared mechanisms between IPF and non-IPF ILD, such as a shared genetic susceptibility [19,20,21,22]. Furthermore, the idea grew that when F-ILD has reached a stage of fibrosis that becomes self-sustaining, antifibrotic treatment could be an effective treatment to slow down disease progression. Because of the similarities in disease behavior and pathophysiological mechanisms with a common downstream pathobiological pathway, the non-IPF progressive F-ILDs (PF-ILDs) have been lumped together, and recently, RCTs also demonstrated the efficacy of antifibrotic treatment in systemic sclerosis-associated F-ILD and a broad range of F-ILDs with a progressive phenotype. [23,24,25,26].

## 3. Immunosuppressive Treatment: Where Are We?

### 3.1. Connective Tissue Disease-Associated ILD (CTD-ILD)

Patients with Sjogren’s syndrome, systemic lupus erythematosus, mixed connective tissue disease, polymyositis and dermatomyositis, rheumatoid arthritis and systemic sclerosis are at risk of developing ILD. Until recently, RCTs have only been performed in patients with systemic sclerosis associated-ILD (SSc-ILD), and evidence from these clinical trials has been extrapolated to other CTDs.

#### 3.1.1. Systemic Sclerosis-Associated ILD (SSc-ILD)

ILD is a common complication in patients with systemic sclerosis and is the leading cause of disease-related deaths [27]. The extent of fibrosis on chest CT is highly predictive of mortality: in a study by Goh et al., patients with extensive disease (extent of fibrosis on CT >30% or 10–30% plus FVC <70% predicted) had a three-fold risk of mortality (hazard ratio (HR) 3.46) compared to patients with limited disease [28]. In 2016, the EULAR (European League against Rheumatism) recommended treatment with cyclophosphamide (CYC) in SSc-ILD as the Scleroderma Lung Study (SLS)-I and Fibrosing Alveolitis in Scleroderma Trial (FAST) showed a beneficial but modest effect of CYC on FVC decline compared to placebo (FVC change in SLS-I: placebo −2.6%, CYC −1.0%; FAST: placebo −3.0%, CYC +2.4%) [29,30,31]. As the treatment effect was mainly due to the prevention of progression and the efficacy of CYC was lost after 24 months, EULAR recommended the use of CYC in particular in patients with progressive disease [31,32]. In the SLS-II study, similar efficacy was seen with two years of mycophenolate mofetil (MMF) compared to one year of oral CYC, but MMF seemed to be better tolerated, especially with less hematological toxicity [33]. Both MMF and CYC resulted in improvements over 24 months in FVC% predicted (FVC%pred). MMF is now considered a first-line therapy for patients with SSc-ILD. Based on observational studies, azathioprine (AZA) can be considered as an alternative treatment when MMF is poorly tolerated [34,35,36,37]. Rituximab, an anti-CD20 monoclonal antibody, is not routinely used as first-line treatment but can be considered in SSc-ILD patients who are progressive despite standard treatment. In a randomized open-label study, rituximab was more efficacious than CYC (change in FVC%pred: rituximab +6.2%, CYC −1.2%) [38].

SSc-ILD has a heterogeneous and variable disease course and close monitoring is important in considering when to start treatment. In a recent published cohort study of 826 SSc-ILD patients, 27% showed progressive ILD (FVC decline >5%) during the initial 12-month period, and 67% experienced progression any time over the mean five-year follow-up [39]. In current clinical practice, treatment is often initiated after progression has occurred, and novel treatment concepts are needed in patients at risk of developing a progressive fibrosing phenotype that aim for the prevention of progression. In this regard, tocilizumab, an IL-6 receptor antibody, is a promising biological agent as exploratory analysis of the faSScinate phase 2 trial and the focuSSced phase 3 trial suggested that tocilizumab could preserve pulmonary function in very early SSc-ILD patients [40,41].

#### 3.1.2. Rheumatoid Arthritis-Associated ILD (RA-ILD)

The development of ILD in RA is associated with a three-times greater risk of mortality, with a median survival of 3–10 years [15,42,43,44,45,46]. Jacob et al. identified 23% of a RA-ILD cohort with an IPF-like progressive fibrotic phenotype based on the combination of the presence of a modified UIP pattern and the presence of extensive fibrosis, which was defined according to the staging system of Goh et al. in SSc-ILD (see above) [28,47]. The management of RA-ILD is challenging, as so far, no RCTs have been performed. Evidence for the use of MMF and CYC has been extrapolated from RCTs in SSc-ILD, as only small cohort studies have been performed in RA-ILD [48,49]. The biologicals rituximab and abatacept, a cytotoxic T lymphocyte antigen 4 (CTLA-4)-immunoglobulin, had a beneficial effect in small open-label and retrospective cohort studies [50,51,52,53,54]. The effect of immunosuppressive agents in RA-UIP remains unclear, and patients with inflammatory types of ILD, such as NSIP and organizing pneumonia, generally have a better response to immunosuppressive agents. There are historical concerns about methotrexate (MTX)-related hypersensitivity pneumonitis and the association between MTX and RA-ILD, but in a recent study by Juge et al., MTX was not associated with an increased risk of RA-ILD, and ILD detection was delayed by 3.6 years [55].

### 3.2. Interstitial Pneumonia with Autoimmune Features (IPAF)

A subgroup of patients that develop interstitial pneumonia have clinical characteristics suggestive of an autoimmune process, but do not fulfill the established criteria for a CTD. Several terms and criteria to describe these patients have been used, including “undifferentiated CTD-associated ILD” (UCTD-ILD), “lung-dominant CTD” or “autoimmune-featured ILD” [56,57,58]. In 2015, the European Respiratory Society (ERS) and American Thoracic Society (ATS) “Task Force on Undifferentiated Forms of Connective Tissue Disease-associated Interstitial Lung Disease” proposed the term “interstitial pneumonia with autoimmune features (IPAF)” and provided classification criteria [59]. So far, no prospective studies have been performed on IPAF. In a retrospective cohort study by Odlham et al., IPAF patients with an UIP pattern had an outcome similar to IPF, while IPAF patients with a non-UIP pattern had an outcome similar to CTD-ILD [18].

### 3.3. Idiopathic Non-Specific Interstitial Pneumonia (iNSIP)

iNSIP has been subcategorized into cellular iNSIP and fibrotic iNSIP, and cellular iNSIP generally responds better to immunosuppressive treatment than fibrotic iNSIP [60,61]. Corticosteroids and other immunosuppressive agents such as AZA, CYC and MMF have always been the cornerstone of management in iNSIP, but no RCTs are available, and only a few retrospective cohort studies have investigated the efficacy of immunosuppressive treatment in iNSIP patients. In a retrospective study by Park et al., the response rate to immunosuppressive treatment—defined as an increase in FVC%pred of ≥10% or diffusing capacity of the lung (DLCO) ≥15% after 1 year—was 53% [62]. Nineteen percent deteriorated despite treatment and had a mortality rate comparable to IPF patients (69% five-year mortality). In a study by Nunes et al., response to treatment was the most robust predictor of mortality [63]. Long-term evolution of pulmonary function tests showed a deterioration in 60% despite immunosuppressive treatment. The response rate to immunosuppressive agents was 26% and was comparable to the previously reported 25% in a study by Danniil et al. and to the reported 29% in a study by Nicholson et al. [64,65]. In summary, the disease course of iNSIP patients is variable, but a proportion of iNSIP patients will deteriorate despite conventional immunosuppressive treatment. 

### 3.4. Sarcoidosis

Sarcoidosis has an unpredictable clinical course ranging from asymptomatic self-limited disease to chronic progressive fibrosing disease that is refractory to treatment [66]. Recently, the American Thoracic Society published first practice guidelines for the diagnosis and detection of sarcoidosis [67]. Walsh et al. provided a simple prognostic staging system for disease staging and for guiding treatment decisions based on the composite physiological index (CPI)—which was developed as a tool by Wells et al. to reflect the extent of fibrosis on CT more accurately than individual pulmonary function test indices—and two HRCT variables: CPI > 40, a main pulmonary artery diameter to ascending aorta diameter ratio of greater than 1 (CPI < 40) or an extent of fibrosis of >20% was highly predictive of mortality (HR 5.89) [68,69]. Most patients do not need any immunosuppressive treatment but in those that do, corticosteroids are considered the cornerstone of treatment [70,71,72]. Second-line therapy includes MTX, leflunomide, MMF and AZA [73,74]. In a retrospective study by Vorselaars et al., MTX and AZA had similar efficacy, but a higher infection rate was observed in the group treated with AZA [75]. TNF-α antagonists are used in cases of refractory disease [76,77]. However, treatment recommendations are based mainly on clinical practice and expert opinion rather than results from RCTs [66].

### 3.5. Hypersensitivity Pneumonitis (HP)

HP is a an immune-mediated disease caused by sensitization to an antigen. Until recently, HP has been classified as acute or chronic HP (cHP) according to clinical presentation [78]. In addition to antigen avoidance, management of HP classically consists of treatment with corticosteroids and other immunosuppressive agents. However, there is only one randomized trial that assessed the efficacy of corticosteroids in a small cohort of patients with acute farmer’s lung. Patients treated with prednisolone had a more rapid improvement of pulmonary function, but after a follow-up of five years, no differences were observed between both groups [79]. De Sadeleer et al. stratified a retrospective cohort of HP patients according to the presence of fibrosis on high-resolution CT (HRCT) and confirmed former evidence that fibrosis correlates more with disease behavior than symptom duration [80]. A beneficial effect of corticosteroid treatment on FVC and DLCO decline was only seen in non-fibrotic HP, but in fibrotic HP, the data suggested a worse outcome. The treatment effect of AZA and MMF was investigated in a retrospective cohort study by Morrisset et al. [81]. Both drugs significantly increased DLCO in cHP patients but had no effect on FVC. No beneficial effect of treatment with corticosteroids before initiation of MMF or AZA was seen in the entire study cohort. Another study demonstrated that the use of MMF or AZA in addition to corticosteroids may decrease adverse events without worsening lung function decline or survival when compared to prednisolone alone [82]. Of note, patients who received immunosuppressive treatment had worsened survival compared to patients who did not. It is unclear whether this increased mortality was the result of a different clinical course or a consequence of immunosuppressive treatment. Recently, guidelines of the American Thoracic Society supported the classification of HP in fibrotic HP and non-fibrotic HP [83]. This might be the start of new research and prospective RCTs in this field.

### 3.6. Occupational ILD

The most common occupational ILDs are asbestosis and silicosis. In fibrosing asbestosis, chest CT resembles the UIP-pattern of IPF, and other radiological features such as pleural thickening or plaques are helpful to make the distinction between IPF and asbestosis. The prognosis is generally better than in IPF, but the disease course depends on its extent and the quantity of asbestosis retained in the lungs [2,84,85]. Simple silicosis is characterized radiologically by small round opacities with an upper-lobe predominance which can coalesce, resulting in progressive massive fibrosis. Some patients present with an UIP-pattern without the typical nodular opacities. The development and progression of chronic silicosis is generally slow, but an accelerated form exists [84,86,87,88,89]. As there are no known immunosuppressive agents that are effective in the treatment of occupational ILD, the mainstay of treatment is avoidance of the causative mineral dusts [84].

### 3.7. Unclassifiable ILD (uILD)

ILD can remain unclassifiable if there are conflicting clinical, radiological or histopathological findings, if the risks of a surgical biopsy do not outweigh the potential benefits or if patients are unable or unwilling to undergo a surgical biopsy [26,90,91,92,93]. Up to 15% of ILD cases are unclassifiable [90]. The prognosis of uILD is variable [92,94]. In a study by Ryerson et al., uILD patients with honeycombing, a UIP or possible UIP pattern, a high fibrosis score and cases in whom a diagnosis of IPF was suspected in the differential diagnosis had an outcome comparable to patients with IPF [92]. However, a proportion of these patients would probably be reclassified as having IPF nowadays, as the current guidelines give more weight to radiological parameters, and recommend a surgical lung biopsy only for cases having a probable or indeterminate UIP pattern on HRCT and for cases with a possible alternative diagnosis [95]. Management of unclassifiable ILD is challenging and should be based on the most probable diagnosis [90,96,97]. There is no direct evidence to guide the decision of immunosuppressive treatment in uILD. However, if the probability of IPF is moderate or high, immunosuppressive agents should be avoided given the increased mortality in IPF patients in the PANTHER-IPF trial [10].

## 4. Antifibrotic Treatment: Where Are We?

### 4.1. Antifibrotics in SSc-ILD

Recently, RCTs provided evidence for the use of antifibrotic treatment in non-IPF (P)F-ILD (Table 1) [98]. SENSCIS, a phase 3 clinical trial, investigated the efficacy and safety of antifibrotics in non-IPF F-ILD [24]. Patients with systemic sclerosis and fibrosis affecting at least 10% of the lungs on HRCT were included. This study demonstrated a reduced FVC decline in patients treated with nintedanib compared to placebo (−52.4 mL vs. −93.3 mL) over a one-year period. An acceptable safety profile of pirfenidone in SSc-ILD patients was demonstrated in an open-label phase 2 study (LOTUSS) [99].

### 4.2. Antifibrotics in PF-ILD

The INBUILD trial included patients with a broad range of non-IPF ILDs [23]. Patients who had fibrotic changes affecting at least 10% of the lungs on HRCT and who were progressive despite conventional treatment were eligible for enrollment. Progression was defined as one of the following criteria within 24 months before screening: (1) a relative decline in FVC%pred of ≥10% or (2) two elements of the following: a relative decline in FVC%pred of ≥5 but <10%, worsening of symptoms or increasing extent of fibrosis on HRCT. This study showed a beneficial effect of nintedanib on FVC decline compared to placebo (−80.8 mL/y vs. −187.8 mL/y). The INBUILD-trial was not powered to provide evidence for the efficacy of nintedanib in the specific ILD entities, but post hoc analysis suggested that nintedanib reduces FVC decline irrespective of the underlying ILD subtype [100]. Results of the RELIEF trial, a phase 2 trial of pirfenidone in non-IPF PF-ILD, so far only have been published in abstract form, but showed a reduction in FVC and DLCO decline in patients treated with pirfenidone compared to placebo [25]. Progression was defined as an annualized absolute decline in FVC%pred of ≥5% within 6–24 months prior to inclusion. A recently published phase 2 trial investigated the efficacy and safety of pirfenidone in progressive uILD [26]. Study participants had F-ILD that was unclassifiable after multidisciplinary discussion and had an absolute decline in FVC%pred of >5% or significant symptomatic worsening six months prior to inclusion, attributable to the underlying uILD. FVC change measured by daily home spirometry was chosen as the primary endpoint, but due to intraindividual variability in this home spirometry, the predetermined statistical model could not be administered to the primary endpoint. However, FVC decline measured by spirometry at study visits, a secondary endpoint, suggested a beneficial effect of pirfenidone in patients with unclassifiable PF-ILD (mean change in FVC at 24 weeks: pirfenidone −17.8 mL vs. placebo −113.0 mL).

Several other clinical trials are planned or ongoing to assess the efficacy and safety of antifibrotics in non-IPF (progressive) F-ILD, as demonstrated in Table 1.

## 5. Unresolved Key Issues in the Pharmacological Management of Non-IPF F-ILD: Where We Would Like to Be 

### 5.1. The Use of Immunosuppressive Agents: The Need for Prospective Studies

Although immunosuppressive treatment has always been the mainstay of treatment in non-IPF ILD, the evidence basis for immunosuppression in non-IPF ILD is limited, as RCTs have mainly been performed in CTD-ILD, and evidence is mainly based on retrospective studies or small open-label studies. Moreover, a substantial proportion of non-IPF ILD patients will deteriorate despite conventional immunosuppressive treatment. Non-IPF ILD patients, and especially those with a progressive fibrosing phenotype, have shared pathogenetic mechanisms with IPF, such as short telomeres: rare genetic variants in telomere-related genes and reduced telomere length have been identified in familial pulmonary fibrosis, sporadic IPF and more recently, in a variety of other ILD subtypes such as cHP, uILD and CTD-ILD [22,101,102,103]. Short telomeres are associated with a progressive phenotype and a poor prognosis [22]. Newton et al. investigated the association between age-adjusted leukocyte telomere length (LTL) and the harmful effect of immunosuppressive agents in the PANTHER-IPF trial [104]. Patients with LTL < 10th percentile that received triple therapy demonstrated a worse composite endpoint-free survival (death, lung transplantation, hospitalization or FVC decline) compared to placebo. In contrast, there was no difference in composite endpoint-free survival for patients randomized to the triple therapy or placebo arms who had LTL > 10th percentile. If this association between telomere length and the effect of immunosuppressive agents is independent of the ILD diagnosis, then it is possible that immunosuppressive agents have a detrimental effect in non-IPF patients with short telomeres. Prospective RCTs are needed to address this question and to identify those patients that benefit from immunosuppressive treatment.

However, immune dysregulation is also thought to be one of the pathobiological mechanisms, especially in CTD-ILD which generally responds better to immunosuppressive treatment than other ILD entities. This highlights the need of pursuing an accurate ILD diagnosis which carries prognostic and therapeutic implications. Moreover, it should be stressed that immunosuppressive treatment improved FVC in RCTs in SSc-ILD, while antifibrotics have only been shown to reduce FVC decline. Moreover, patients in the SLS-II trial had worse pulmonary-function baseline characteristics compared to patients in the SENSCIS trial [24,33]. Several clinical trials investigating the efficacy of immunosuppressive agents and biologicals are still ongoing (Table 2). Combination treatment with MMF and rituximab is currently being investigated in patients with iNSIP and NSIP associated with CTD or IPAF that do not respond to first-line immunosuppressive treatment (EvER-ILD (NCT02990286)), and a head-to-head comparison between rituximab and CYC is currently being performed in patients with severe and/or progressive CTD-ILD (RECITAL (NCT01862926)) [105,106]. The PULMORA-trial (NCT04311567) will investigate the role of tofacitinib in RA-ILD, and abatacept and basiliximab are currently being investigated in myositis-associated ILD (ATtackMy-ILD-NCT03215927)/RA-ILD (APRIL-NCT03084419) and amyopathic dermatomyositis-associated ILD (NCT03192657), respectively.

### 5.2. The Use of Antifibrotics: The Need for Adequately Defining Disease Progression

Antifibrotics have been shown to be effective in large, good-quality RCTs on PF-ILD, but to date there is no universally accepted definition of progression [23,25,26]. The eligibility criteria of clinical trials have provided a starting point to establish standardized criteria for disease progression. These criteria are based on the occurrence of one or more of the following recognized clinical parameters of progression within a certain time period: worsening of respiratory symptoms or lung function and increasing extent of fibrosis on HRCT. The duration of the time period and thresholds for defining “worsening” and “increasing extent” are not standardized and vary between studies. Another concern is that all definitions of progression imply that patients should wait for clinical deterioration and loss of lung capacity before a progressive fibrosing phenotype can be designated. Further research to identify and validate predictors of progression is needed with the aim of initiating antifibrotic treatment before progression has effectively occurred.

Furthermore, it is worth nothing that antifibrotic treatment is much more expensive than the existing immunosuppressive agents and should therefore be used in a well-selected population (e.g., PF-ILD patients) to guarantee the cost-benefit ratio.

### 5.3. The Choice between Antifibrotics, Immunosuppressive Agents and Combination Treatment: The Need for Biomarkers

In the future, pharmacological treatment options will consist of no pharmacological treatment, immunosuppressive agents or antifibrotics or even a combination of these treatments. The question as to which drug(s) to use in a particular patient has not been answered yet, as no head-to-head comparisons have been performed. However, soon, clinicians will need to make the decision regarding which therapy to initiate. The identification and validation of biomarkers such as clinical features and genetic and molecular profiles that reflect upregulated pathways, predict disease behavior and identify patients at increased risk of harmful effects of treatment may help to initiate appropriate treatment early in the disease course. Based on current knowledge, it appears likely that patients with an UIP pattern on histopathology or CT or patients with extensive fibrotic disease will benefit from antifibrotic treatment. However, it should be stressed that the INBUILD trial and RELIEF trial selected patients who were progressive despite conventional immunosuppressive treatment and that antifibrotic drugs as a first-line treatment in non-IPF PF-ILD has not yet been investigated.

In the near future, combination treatment with immunosuppressive agents and antifibrotics will be a potential option. The tolerability and safety of combination treatment has been demonstrated in SSc-ILD, but the role of antifibrotic treatment as add-on treatment has not yet been investigated. In the SENSCIS trial, the use of immunosuppressive agents was not randomized, but the FVC decline in the placebo group was lower in patients receiving MMF, suggesting a potential beneficial effect of MMF in fibrosing SSc-ILD. Moreover, patients receiving both therapies had the slowest rate of FVC decline (MMF + nintedanib: −40.2 mL/y; MMF + placebo: −66.5 mL/y; nintedanib: −63.9 mL/y; placebo: 119.3 mL/y), suggesting a potential role for combination treatment in fibrosing SSc-ILD [24].

However, the difference between patients receiving MMF + placebo and those receiving MMF + nintedanib was only small (26.3 mL) and it is unclear whether this small difference is clinically significant. The safety of combination therapy with pirfenidone and MMF has also been demonstrated in progressive fibrosing uILD, but assessment of the effect of MMF on FVC decline was not possible in this study due to small sample sizes [26]. The SLS-3 trial (NCT03221257) is ongoing and investigates the role of combination treatment in SSc-ILD: 150 patients will be randomized to MMF plus pirfenidone or MMF monotherapy, with the primary outcome being the change in FVC%pred from baseline over a time period of 18 months.

## 6. Conclusions

The clinical course in patients with PF-ILD mimicks the relentless progressiveness of idiopathic pulmonary fibrosis (IPF) and recently, RCTs demonstrated the efficacy of antifibrotic treatment in non-IPF ILD. However, several questions remain and need to be addressed in future studies: which patients benefit from immunosuppressive therapy? Should antifibrotics be restricted to PF-ILD or should it be used in any F-ILD patient? What is the role of combination treatment? Which antifibrotic and immunosuppressive treatment can be combined safely? Further research is needed to address these questions and to identify and validate biomarkers that will help to initiate appropriate treatment early in the disease course.

## Figures and Tables

**Table 1 jcm-10-01330-t001:** Randomized clinical trials with antifibrotics among patients with (progressive) fibrosing ILD.

Study/Phase Study Identifier	Patients (*n*) Duration (Weeks)	Criteria for Defining Fibrosis and Progression	Use of Immunosuppressive Agents	Primary Outcome
**Published RCTs**				
Ssc-ILD				
SENSCIS: Nintedanib in SSc-ILD [24] Phase 3 (NCT02597933)	*n* = 576 52 weeks	- fibrosis: fibrosis affecting >10% of the lungs on HRCT - progression:/	A stable dose of prednisone (<10 mg/day), MMF and/or MTX for at least six months before randomization was permitted.	Annual rate of FVC decline
LOTUSS: Pirfenidone in SSc-ILD [99] Phase 2 (NCT01933334)	*n* = 63 20 weeks	- fibrosis:/ - progression:/	Stable doses of oral CYC or MMF for at least three months before randomization was permitted.	% patients with TE AEs or TE SAEs
PF-ILD				
INBUILD: Nintedanib in PF-ILD [23] Phase 3 (NCT02999178)	*n* = 663 52 weeks	- fibrosis: reticular abnormality with traction bronchiectasis with or without honeycombing, with disease extent of >10% on HRCT - progression: one of the following criteria within 24 months before screening despite standard treatment: • Relative decline in FVC%pred of ≥10% • 2 elements of the following: relative decline in FVC%pred ≥5% but <10%, symptom worsening or an increasing extent of fibrosis on HRCT	Glucocorticoids <20 mg/day was permitted.	Annual rate of FVC decline
RELIEF: pirfenidone in PF-ILD [25] Phase 2 (DRKS00009822)	*n* = 127 48 weeks	- fibrosis: fibrotic lung disease on HRCT - progression: annualized absolute FVC decline ≥5% within 6–24 months before screening	A stable dose of prednisolone (≤15 mg/day) or stable dose of other immunosuppressive therapy for at least three months before randomization was permitted.	Absolute change in FVC%pred from baseline
Progressive uILD				
pirfenidone in uILD Phase 2 (NCT03099187)	*n* = 253 24 weeks	- fibrosis: fibrosis affecting ≥10% of the lungs on HRCT - progression: One of the following criteria within six months before inclusion: • absolute decline in FVC of > 5%pred • significant symptomatic worsening not due to cardiac, pulmonary, vascular or other causes.	A stable dose of MMF for at least three months before randomization was permitted.	FVC decline assessed by daily home spirometry during the 24-week study period
**Planned and ongoing clinical trials in non-IPF F-ILD**				
CTD-ILD				
SLS-III: Combining pirfenidone with MMF in SSc-ILD Phase 2 (NCT03221257)	*n* = 150 18 months	- fibrosis:/ - progression:/	Prednisone <10 mg/day and the use of MMF is permitted if the responsible physician indicates that continued use is in the best clinical interest of the patient.	Change in FVC%pred from baseline
Pirfenidone in SSc-ILD Phase 3 (NCT03856853)	*n* = 144 52 weeks	- fibrosis:/ - progression:/	Treatment with prednisone <10 mg/day and stable doses of MMF and/or MTX for at least six months before inclusion is permitted.	Relative change in FVC% from baseline
TRAIL-1: pirfenidone in RA-ILD Phase 2 [100] (NCT02808871)	*n* = 270 52 weeks	- fibrosis: reticular abnormalities affecting >10% of the lungs on HRCT - progression:/	Prednisone <20 mg/day and a stable dose of immunosuppressive agents for at least three months before randomization is permitted.	incidence of the composite endpoint of decline in FVC%pred of >10% or death
Pirfenidone in progressive ILD associated With Clinically Amyopathic Dermatomyositis Phase 4 (NCT02821689)	*n* = 57 52 weeks	- fibrosis:/ - progression: increase in level of dyspnea and worsening of fibrosis on HRCT with >10% increase of HRCT score and/or absolute decrease in FVC%pred of >10% within 3–6 months after diagnosis.	Patients ever treated with biologicals, including basiliximab, were excluded.	12-month survival from the onset of ILD
Pirfenidone in DM-ILD Phase 3 (NCT03857854)	*n* = 152 52 weeks	- fibrosis:/ - progression:/	Stable dose of prednisolone <15 mg/day for more than one month and of other immunosuppressive agents for more than three months before inclusion is permitted.	Relative change from baseline of FVC
Sarcoidosis				
PiRFS: Pirfenidone for Progressive Fibrotic Sarcoidosis Phase 4 (NCT03260556)	*n* = 60 104 weeks	- fibrosis: fibrosis affecting >20% of the lungs on HRCT - progression:/	A stable dose of prednisone <20 mg/day and other immunosuppressive agents for at least two months before inclusion is permitted.	Time until clinical worsening
HP				
Pirfenidone in Fibrotic HP Phase: NA (NCT02958917)	*n* = 44 52 weeks	- fibrosis: reticular abnormality, and/or traction bronchiectasis, and/or architectural distortion and/or honeycombing on HRCT - progression:/	Not specified.	Mean change from baseline in %FVC

F-ILD: fibrosing interstitial lung diseases; PF-ILD: progressive fibrosing ILD; SSc-ILD: systemic sclerosis-associated interstitial lung disease; cHP: chronic hypersensitivity pneumonitis; DM-ILD: dermatomyositis-associated interstitial lung disease; RA-ILD: rheumatoid arthritis-associated interstitial lung disease; uILD unclassifiable interstitial lung disease; MMF: mycophenolate mofetil; MTX: methotrexate; CYC cyclophosphamide; TE AEs: treatment-emergent adverse events; TE SAEs: treatment-emergent severe adverse events; HRCT: high-resolution computed tomography; FVC: forced vital capacity.

**Table 2 jcm-10-01330-t002:** Ongoing and unpublished randomized clinical trials with immunosuppressive agents in CTD-ILD.

Study/Phase Study Identifier	Patients (n) Duration (Weeks)	Criteria for Defining CTD, ILD and Severity/Progression	Use of Immunosuppressive Agents	Primary Outcome
PULMORA Effects of Tofacitinib vs. MTX on RA-ILD Phase 4 NCT04311567	*n* = 48 24 weeks	- CTD: diagnosis of seropositive RA within 24 months prior to inclusion - ILD: pulmonary abnormalities suggestive of RA-ILD -severity/progression:/	Previous treatment with DMARDs is not allowed. History of prednisone use is allowed but should have been discontinued two weeks before baseline visit.	Change in total interstitial disease score of pulmonary abnormalities by HRCT
APRIL AbatacePt in RA-ILD Phase 2 NCT03084419	28 weeks *n* = 30	- CTD: diagnosis of RA - ILD: ILD associated with RA with supportive findings on PFT and HRCT - Progression: a decrease in FVC >5% within 24 months prior to inclusion or progression of lung fibrosis on HRCT as reported by a chest radiologist.	Treatment with other immunosuppressive agents, e.g., MMF—unless this has been discontinued with an adequate washout period—is not allowed. A stable dose of MTX and hydroxychloroquine for at least six weeks prior to baseline visit is permitted. Treatment with >10 mg prednisolone daily within six weeks or rituximab within 24 weeks prior to baseline visit is not allowed.	Change in FVC
SCLEROCYCCYC in SSc-ILD Phase 3 NCT01570764	*n* = 40 52 weeks	- CTD-ILD: SSc-ILD - Progression: worsening of ILD on HRCT and worsening of FVC and/or TLC ≥10% and/or worsening of DLCO ≥ 15% as compared to values obtained within 3–18 months preceding inclusion	Treatment with CS >15 mg/d during the last three months, CYC during the last 12 months or rituximab during the last six months prior to inclusion is not allowed. Treatment with MTX or MMF at inclusion is not allowed.	Change in FVC
EvER-ILD Evaluation of Efficacy and Safety of Rituximab With MMF in Patients With ILD [105] Phase 3 NCT02990286	*n* = 122 26 weeks	- (CTD-)ILD: diagnosis of NSIP on histopathology or HRCT (basal predominant reticular abnormalities with traction bronchiectasis, peri-bronchovascular extension and subpleural sparing, frequently associated with ground-glass attenuation) and associated with differentiated CTD of IPAF or idiopathic ILD - Progression: patients who did not respond, relapsed or were not able to continue at least one first-line immunosuppressive treatment of ILD. The absence of response was defined as either a decrease or an increase, but < 10% in FVC%pred.	Treatment with immunosuppressive agents other than CS (AZA, CYC, MTX, cyclosporine, tacrolimus, leflunomide) within two weeks prior to inclusion or IVIG, hydroxychloroquine or other monoclonal antibody therapies within six months prior to inclusion are not allowed.	Change in FVC%pred
RECITALR ituximab Versus CYC in CTD-ILD [106] Phase2/3 NCT01862926	*n* = 116 48 weeks	- CTD-ILD: ILD associated with SSc, idiopathic interstitial myopathy and MCTD-severe and/or progressive ILD: intention of the caring physician to treat the ILD with IV CYC (deteriorating symptoms attributable to ILD, deteriorating PFTs, worsening gas exchange or extent of ILD) and when there is a reasonable expectation that immunosuppressive treatment will stabilize or improve CTD-ILD. In individuals with SSc, it is anticipated that subjects will fulfil the criteria for extensive disease defined by Goh et al.	Immunosuppressive therapy (other than CS) received within two weeks prior to inclusion is not allowed. Previous treatment with rituximab and/or intravenous CYC is not allowed.	Change in FVC
ATtackMy-ILD Abatacept in Myositis-associated ILD Phase 2 NCT03215927	*n* = 20 24 weeks	- CTD: diagnosis of anti-synthetase syndrome - ILD: reticulation, honeycombing or ground glass opacities (GGO) - Severe and/or progressive ILD (a) baseline FVC <80% or (b) FVC 80–100% with ≥10% decline in FVC in the last 12 months prior to inclusion	Inclusion criterium is the use of a stable dose of steroids, one of the other immunosuppressive agents (MMF or AZA) or a combination of steroid and an immunosuppressive agent. Other immunosuppressive agents, including MTX, cyclosporine, IVIG, tacrolimus, CYC or tofacitinib, are not allowed. Biologicals, i.e., rituximab, anti-TNF agents, tocilizumab, are not allowed.	Change in FVC%pred
CATR-PAT CYC and AZA vs. Tacrolimus in Anti-synthetase Syndrome-related ILD Phase 3 (open label) NCT03770663	*n* = 76 52 weeks	- CTD-ILD: ILD associated with anti-synthetase syndrome - Moderate to severe ILD: FVC <80% and/or DLCO <70%	Previous use of CYC, AZA or tacrolimus in the last six months prior to inclusion is not allowed. Previous use of three daily IV steroids <3 months before inclusion is not allowed. Patients with worsening or relapse under prednisone >0.5 mg/kg/day are excluded.	Progression-free survival
Basiliximab as a Treatment of Interstitial Pneumonia in Clinical Amyopathic Dermatomyositis Patients Phase 2NCT03192657	*n* = 100 52 weeks	- CTD-ILD + severity/progression: Dermatomyositis and interstitial pneumonia and meeting at least two of four criteria: (1) interstitial pneumonia images on HRCT, (2) DLCO ≤ 60%, (3) elevated serum KL-6, (4) elevated serum anti-MDA5 (+).	Previous application of immunosuppressive agents or any target treatment for dermatomyositis is not allowed.	Survival

ILD: interstitial lung disease; CTD-ILD: connective tissue disease-associated ILD; NSIP: non-specific interstitial pneumonia; IPAF: idiopathic pneumonia with autoimmune features; SSc-ILD: systemic sclerosis-associated interstitial lung disease; RA-ILD: rheumatoid arthritis-associated interstitial lung disease; MCTD: mixed connective tissue disease; CS: corticosteroids; MMF: mycophenolate mofetil; MTX: methotrexate; DMARD: disease modifying anti-rheumatic drugs; CYC cyclophosphamide; AZA: azathioprine; HRCT: high-resolution computed tomography; PFT: pulmonary function test; FVC: forced vital capacity; DLCO: diffusing capacity for carbon monoxide; TLC: total lung capacity; IV: intravenous; IVIG: intravenous immunoglobulin.

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
