# Peer review of "Progression in the Management of Non-Idiopathic Pulmonary Fibrosis Interstitial Lung Diseases, Where Are We Now and Where We Would Like to Be"

_jcm, 2021, doi:10.3390/jcm10061330_

Round 1
Reviewer 1 Report
I was disappointed to see that this review is very biased, systematically downplaying the efficacy and safety of immunosuppressive therapy, and focusing on the (very questionable) efficacy and safety of nintedanib, which is still to be demonstrated, and far from being established.
The authors are writing a review article and should have been much more objective and balanced when considering and discussing these data.
I explained the concerns over nintedanib data below, but the authors even managed to promote the use of pirfenidone, for which the evidence is basically non-existing at this point, with misleading statements.
If the average, non-expert reader read this, they will get the impression that we are harming patients with immunosuppressive therapy (as we were in IPF in the past). This is definitely not the case. In fact the “questionable quality” they quote applies to the recent studies on anti-fibrotics in non-IPF ILD.
Starting from the abstract, and then in the main text, the authors make several misleading statements, unfortunately: “since immunosuppressive treatment in non-IPF ILD is very limited” – yes there are only a few trials, but they all unanimously show positive results, and much better than those on nintedanib at this point of time.
“the deleterious effect of immunosuppression in IPF patients has been demonstrated” – we are not talking about IPF here, and the “deleterious” effect cannot be translated to a different group of conditions.
“In contrast, randomized placebo-controlled trials demonstrated the efficacy and safety of pirfenidone (??) and nintedanib in non-IPF progressive fibrosing ILD and systemic sclerosis associated-ILD.” – I invite the authors to review the evidence more objectively:
- No comparison with standard immunosuppressive therapy was made.
- Nintedanib was not tested as add-on therapy.
- “Progressive disease” was not considered in the SSc trial, and very questionable in the other trial, where the worsening on HRCT was not specified in any way and a decline as little of 5% could represent so-called “progression”; all this made their definition of “progression” very questionable and arbitrary.
- Patients in the SSc trial had milder disease than in the MMF-cyclophosphamide trial (Tashkin et al), yet MMF/cyclo were more efficacious, not only stabilizing, but even improving lung function.
- In the SSc trial with nintedanib, in the subgroup of patients who were actually on MMF, no significance difference was seen.
- Whether the small difference shown in FVC decline is clinically significant, it remains to be established.
- Side effects of nintedanib were numerous and very concerning, such as high incidence of nausea and vomit in pts with SSc, which could promote further aspiration.
- Nintedanib is much more expensive than the existing therapies.
Re pirfenidone: I invite the authors to read the data more carefully as well – there is really no evidence at all supporting its use in so-called “unclassifiable ILD”: the trial did not even meet the main endpoint! And yet the authors stated its presumed “efficacy”.
The introduction on immunosuppressive therapy in IPF missed the point – again, we are not talking about IPF here, and the authors are giving the impression that these therapies are destined to fail, and this is misleading.
In the scleroderma therapy section, I’m disappointed to read how the authors downplay the results of Tashkin et al., who showed significant and sustainable improvement (not just stabilization or less decline!) of lung function. In fact, these are not even mentioned at all.
There are also several recent and good quality studies on immunosuppressive therapies that the authors did not mention or discuss: Resp Med 2016;121:117-122; Pulm Pharmacol Ther 2020;60:101878; Boerner et al. Respiration 2020 Jul 14;1-9.
Given that all published studies unanimously showed positive results, I really don’t understand how the authors could make a statement such as “Based on the current literature, the role of immunosuppressive agents might be overestimated in iNSIP, especially in the fibrotic forms”. This is not supported by evidence.
Sarcoidosis section: please explain what the composite physiologic index is.
Unclassifiable ILD: “patients with an UIP pattern and honeycombing” – this is IPF by definition and would obviously be treated with anti-fibrotics. There is confusion here being made between truly unclassifiable ILD and IPF diagnosed based on clinical-radiographic criteria only.
I realize that this confusion was made by the quoted article rather than by authors, but it does add confusion to the reader of the present review.
Antifibrotics in PF-ILD: the difference in decline was actually -81 vs -188 cc. The authors reported results only on patients with “UIP-like patterns” and this is again misleading.
“favorable effect is mainly driven from the non-fibrotic cases or is limited to short-term outcome” again a misleading statement, look at the baseline function in Tashkin et al vs. Senscis for example.
“potentially harmful effect of immunosuppression in non-IPF ILD considering the PANTHER-IPF-trial and shared pathogenetic mechanisms with IPF such as short telomeres”: this is frankly not acceptable in a scientific article. The authors are confusing IPF and non-IPF ILD and the evidence supporting this statement is non-existing.
“clinicians will need to make the decision which therapy to initiate based on the presence of either inflammation and/or fibrosis as the main pathogenetic driver” – that’s oversimplistic: inflammation and fibrosis coexists in many types of ILD without this necessarily predicting a response to immunosuppressive vs anti-fibrotic therapy. A predictable response may be dictated underlying processes such as cell death, senescence, innate or adaptive immune reactions, etc.
“Patients with non-IPF PF-ILD have a disease course comparable to IPF.” The authors should be more careful before making such statement. There isn’t a single, long term perspective study that has ever showed that, and generalizing can be misleading.
I’m not sure if the authors fully understand that all these pro-antifibrotic statements not supported by adequate evidence can also have a negative impact from a diagnostic perspective: by encouraging to prescribe anti-fibrotics to every “progressive ILD” no matter what, the specialist may be pushed away from pursuing an accurate diagnosis of the interstitial process, which is exactly the opposite of what an ILD expert should do.
Reviewer 2 Report
Reviewers' comments for the manuscript “Progression in the management of interstitial lung diseases, where are we now and where we would like to be”
This review article provides an overview of the historical background, current knowledge, and prospects of the treatment of F-ILD. The content is easy to understand for beginners and will be beneficial for experts. The ongoing clinical trials are also well summarized and informative. Although there are some negative comments about immunosuppressive therapy, I think this is acceptable considering the INBUILD trial's excellent results.
I have some minor comments about this article.
- I feel uncomfortable that pirfenidone is listed together with nintedanib in the abstract. The efficacy of pirfenidone has only been reported in two phase II trials. Moreover, one is only available as an abstract and another did not meet the primary outcome. Although I agree with its effectiveness, I think it should be distinguished from nintedanib on abstract.
- This review article is poorly described regarding PM/DM related ILD. 41% of PM/DM patients have been reported to be complicated by ILD (Sun et al. Semin Arthritis Rheum. 2020). PM/DM-related ILD is serious problem, and has attracted the interest of many respiratory physicians and rheumatologists. Some prospective studies have reported the effects of immunosuppressive treatment for PM/DM-related ILD (e.g., Fujisawa et al. Respirology. 2020 and Takada, et al. Rheumatology. 2020). Please consider adding a description of PM/DM related ILD.
- "If this association between telomere length and the effect of immunosuppressive agents is independent of the ILD diagnosis, then it is possible that immunosuppressive agents have a detrimental effect in non-IPF patients with short telomeres." Page7, Line280.
This sentence is complicated and confusing for me. Please consider making it easier to understand.
I hope you will find the comments useful.
Reviewer 3 Report
This review article well summarized the current researches and gives readers a perspective of the future research.
#1 P1 Title and abstract
The title – In the management of ILD- is too broad to define the main idea of this article. The abstract is about non-IPF ILD, and mainly describes progressive fibrosing non-IPF ILD. Please more clarify and specify the title compatible with the abstract.
#2 P2 L48
randomized clinical trials-> randomized controlled trials (RCTs)
#3 P2 L68-70
RCTs also demonstrated the efficacy of antifibrotic treatment (i.e. pirfenidone and nintedanib) in PF-ILD and in systemic sclerosis associated-ILD (SSc-ILD)
- This expression seems to that PF-ILD and SSc-ILD are different subset, so some readers may get confused that PF-ILD and SSC-ILD are different subsets. In fact, PF-ILD includes SSc-ILD, so I suggest revise the expression to clarify the disease hierarchy.
#4 P3 L114
(cf. supra)
à what is the exact meaning?
# P3 L126
3.1.3 Interstitial pneumonia with autoimmune features (IPAF)
à IPAF is a suggested concept for standardizing of the terms and definition for future research, not for clinical application yet. As authors described, these patients do not fulfill the established criteria for a CTD. For example, a patient with UIP CT pattern/IPAF, we still treat the patent as a IPF patient according to current guideline. Therefore, the arrangement of this IPAF section in the subcategories of CTD-ILD seems to be inappropriate.
#P4 L155
3.3 Sarcoidosis
à Recently, ATS published first practice guidelines for diagnosis and detection of sarcoidosis. Although this review focuses on the treatment, if possible, citing the recent guideline can help readers get new information related to the disease. (Authors also describe new classification of HP in the next section)
# P5 L194
3.5. Exposure-induced ILD
à I understand the intention of authors for using the term ‘exposure-induced ILD’, but this term is not commonly used. In addition, it seems to be not specific for occupational/environmental lung diseases, for example, it may be used for other conditions, such as radiation exposure, dust exposure, and so on. In this section, authors mostly described occupational diseases. I suggest the term of environmental/occupational lung disease as an alternative option, Instead of exposure-induced ILD.
# P5 L207
3.6. uILD
à The refence 26 (DOI:https://doi.org/10.1016/S2213-2600(19)30341-8) is the first clinical trial for the treatment of progressive uILD, so please cite this article in the section of u-ILD as well.
# P5 L223
SENSCIS was the first phase 3 clinical trial that investigated the 223 efficacy and safety of antifibrotics in non-IPF F-ILD [24].
- INBUILD trial was published earlier than SENSIS trial, and INBUILD trial also investigated antifibrotics in non-IPF PF-ILD. Please edit this sentence.
# P6 L254
Table 1
- Please cite the refence numbers in the published studies in Table 1.
# P10 L355
If authors can revise the conclusion section more concisely, it is helpful to clarify the main idea of this article.
